# Artificial Weathering Mechanisms of Uncoated Structural Polyethylene Terephthalate Fabrics with Focus on Tensile Strength Degradation

**DOI:** 10.3390/ma14030618

**Published:** 2021-01-29

**Authors:** Hastia Asadi, Joerg Uhlemann, Natalie Stranghoener, Mathias Ulbricht

**Affiliations:** 1Institute for Metal and Lightweight Structures, University of Duisburg-Essen, Universitaetsstr. 15, 45141 Essen, Germany; joerg.uhlemann@uni-due.de (J.U.); natalie.stranghoener@uni-due.de (N.S.); 2Lehrstuhl für Technische Chemie II, University of Duisburg-Essen, Universitaetsstr. 7, 45117 Essen, Germany; mathias.ulbricht@uni-essen.de

**Keywords:** PVC-coated PET woven fabric, uncoated PET woven fabric, weathering mechanisms, artificial weathering

## Abstract

In the past five decades, reinforced coated textile membranes have been used increasingly as building materials, which are environmentally exposed. Thus, their weathering degradation over the service life must be taken into account in design, fabrication, and construction. Regarding such structural membranes, PVC (polyvinylchloride)-coated PET (polyethylene terephthalate) fabric is one of the most common commercially available types. This paper focuses on the backbone of it, i.e., the woven PET fabric. Herein, weathering of uncoated PET, as the load-bearing component of the composite PET-PVC, was studied. This study assessed the uniaxial tensile strength degradation mechanisms of uncoated PET fabric during artificial accelerated weathering tests. For this purpose, exploratory data analysis was carried out to analyze the chemical and physical changes which were traced by Fourier transform infrared spectroscopy and molecular weight measurements. Finally, with the help of degradation mechanisms determined from the aforementioned evaluations, a degradation pathway network model was constructed. With that, the relationship between applied stress, mechanistic variables, structural changes, and performance level responses (tensile strength degradation) was assessed.

## 1. Introduction

Architectural membrane materials consist of a polymeric matrix with fibers as both flexible and extensible cladding and reinforcement. Among two different reinforcement covering mechanisms (laminates or fabrics coated), coated woven fabrics have been widely used due to the good weathering resistance and less possibility of delamination. There are single- or multi-component coated fabrics [1]. The selection of a membrane for structural purposes has various criteria such as mechanical properties (tensile strength, breaking strain, elastic features, etc.), insulation, light transmission, fire retardancy, foldability, and cost [1]. Durability assessment is one factor for reducing costs, which is quantified by a material’s useful lifetime. This helps manufacturers to plan maintenance and replacement procedures in advance [2]. For improving the durability, the ageing process (a complex phenomenon that causes mainly unfavorable changes over the material’s service live) should be scrutinized. The most accurate results come from outdoor exposure, but this evaluation is very slow. Therefore, accelerated weathering techniques have been used, with a combination of some important damaging factors as UV radiation, oxygen, temperature, and humidity [2]. Mimicking other factors such as air pollution and microorganisms requires new techniques.

In this paper, firstly, an overview is provided on the structure of PET (as the backbone of PET-PVC architectural fabrics), its weathering mechanisms under various environmental stressors and the consequences for PET stability and integrity (Section 2). Secondly, the sensitivity of tensile strength to each stressor is examined by performing artificial weathering tests with different combinations of UV, humidity, and temperature (Section 3 and Section 4). Finally, possible mechanisms of tensile strength degradation are evaluated and discussed by monitoring chemical and physical structural changes of PET at each exposure period with the help of two techniques, infrared spectroscopy and viscosity tests (Section 4). Here, an attempt is made to illustrate the relationship between weathering mechanisms (direct or indirect influence on the tensile strength) and measured evidence, by exploratory data analysis.

## 2. Overview on the State of the Knowledge

### 2.1. PVC-Coated PET Woven Fabric and Its Weathering Behavior

PET-PVC is a kind of translucent multi-component coated woven fabric. In this composite, woven PET fabric is stabilized and protected by main and top coatings. Figure 1 depicts the different layers of this composite [3].

Under weathering conditions, first the topcoat blisters and gradually disappears [4]. A TiO_2_ containing primer can provide UV stability. After the extinction of topcoat and primer, the PVC main coat is exposed to weathering impacts. This PVC main coating is comprised of the polymer PVC itself, softener, thermo-stabilizer, and organic UV-absorber. In this composition, weathering resistance is provided by stabilizer and UV-absorber. With the passage of time, softeners (plasticizers) are prone to migrate to the surface and evaporate [5]. At this stage, aging cracks appear on the brittle PVC surfaces. Micrographs taken from two realized projects are shown in Figure 2, illustrating that the weathering impacts could alternate the coating surface with bulges and/or pulverization (consistent with [6]) or, in severe condition after a long time, could even wash out coatings completely. For example, in Figure 2b, coating decomposes to a degree where the yarns lie almost bare after 38 years. João et al. [7] defined a degradation path on PET-PVC specimens, which starts by surface soiling, and leads to darker surfaces absorbing more heat. Resulting high temperature accelerates chemical aging where consequently softener evaporates, and material gets brittle. In this way, small cracks appear (see Figure 2a) and, finally, PET fibers are directly affected by weathering conditions.

Considering these damages of PVC-based coatings under harsh weathering situations or local damages such as abrasion, some PET fabrics might not just be partially covered by coating layers but completely unprotected. Hence, the PET itself is left vulnerable in the end. Considering that the uniaxial tensile strength is mainly provided by the PET woven yarns in the composite structure, this state at the latest marks the end of the useful lifetime of the textile membrane. Therefore, weathering impacts on the tensile strength should be traced in structures of PET filaments.

### 2.2. Mechanisms of PET Degradation under Influence of Different Factors

PET is a thermoplastic semicrystalline material with a glass transition temperature between 67 and 80 °C. The macromolecular structure comprises zigzag chains, connecting to each other by van der Waals forces [8]. In this structure, the phenyl group (see Figure 3) hinders the rotation of the chain segments, which causes high stiffness, while the short ethylene group increases flexibility [9,10].

Degradation is a chemical process, which influences both chemical and physical properties such as chain conformation, molecular weight distribution, chain flexibility, chain crosslinking and branching, crystallinity, and color. Various forms of PET degradations are photodegradation, photooxidation, hydrolysis, thermal degradation, and chemical degradations, but typically combinations of impacts and effects are observed. The degradation mechanisms lie among three extreme cases: random chain scission, depolymerization, and disappearance of entire macromolecules from the solid polymer [11]. In random chain scission, no volatile material is produced, while, in the second mechanism, volatile materials are progressively separated from the chain ends. In this way, the decrease of the molecular weight is proportional to the amount of volatile products [12]. 

The most significant weathering factors responsible for PET degradation are oxygen, UV light, moisture, and temperature, as discussed subsequently below.

#### 2.2.1. Atmospheric Oxygen

When oxygen molecules attack a hydrocarbon, a hydroperoxide group is formed first. Hydroperoxides play a major role in chain scission because of the weak O-O bond, favoring dissociation to yield radicals ([13,14,15]) (see Reaction 4 in Figure 4) [16]. Oxygen also accelerates crosslinking by Reactions 8–10 in Figure 4. Presence of oxygen influences other degradation mechanisms, such as thermal oxidation and photooxidation, which result in rapid formation of hydroperoxides and rapid chain scission [17] (see also below). Chain scission leads to mobile small chains which could rearrange and crystallize (chemi-crystallization). This effect has been seen during oxidative degradation [16].

#### 2.2.2. UV Light

Studies of photodegradation for PET go back to the early 1960s. Photodegradation is initiated by UV absorption and the effects are strongly influenced by oxygen. Light in the 290–400 nm wavelength range of sunlight can cause photolysis of PET [11]. Malanowski [18] believed that the consequences of this process might be restricted as radicals are formed at the surface and react with oxygen (Reactions (1) and (2) in Figure 4). This chemically trapped oxygen may cause a lack of required oxygen inside the material thickness [19]. In fact, when the photochemical reaction near the surface proceeds so rapidly, oxygen moleculesare used up before they can diffuse very deeply into the samples. By cutting off light sources (nights during natural weathering or when the light sources switch off during artificial weathering tests), the rate of O_2_ absorption decreases to stationary rate and oxidation reaction proceeds for some time. This effect is called “photochemical after effect” or “post-effect” [20]. It should be noted that PET yarns, which are utilized for membrane fabrics, are not pigmented and therefore sensitive to UV radiation [8].

Photooxidation mechanisms of organic polymers, with hydrocarbon structure, generally consist of three steps [21]. In the following explanations, h is the Planck’s constant and ν is the frequency of radiation. When a molecule absorbs electromagnetic radiation, its energy increases by the amount of energy of absorbed photon, E = hν:Initial step:

Formation of free radicals by the following equation,
$$\mathrm{RH}\;(\mathrm{polymer})\overset{h\nu \left(O_{2}\right)}{\to }R{}^{\circ}(\mathrm{polymeralkyl}\;\mathrm{radical})+\mathrm{HO}{}^{\circ}(\mathrm{hydroperoxy}\;\mathrm{radical})$$

2.Propagation steps:

Reaction of free radicals with oxygen and producing polymer hydroperoxide radicals, which can lead to chain scission (see Reactions 2–4 in Figure 4). Philip and Al-Azzawi [13] believed the chain scission process in natural weathering is slower than in accelerated weathering.

3.Termination steps:

Reaction of different free radicals resulting in crosslinking (bimolecular combination) (see Reactions 8–10 in Figure 4). Bimolecular termination of ROO° probably occurs by a six-membered intermediate such as Norrish Type II (intermediate step).

The light-absorbing parts of the PET molecular structure, ester carbonyl group and aromatic group, are responsible for photochemical events [19]. The photochemical degradation mechanisms can be differentiated into Norrish I and II, as shown in Figure 5.

In the Norrish Type I reaction, three types of radicals are generated: carboxyl, phenyl, and carbonyl. Carboxyl radical can either form a carboxylic acid end group by abstraction of hydrogen from neighboring polymer (1–8) or release CO_2_ to form a phenyl radical (1–4) (cf. Figure 5). As another alternative, carbonyl radical abstracts hydrogen to form aldehydic chain end (3–9) or generates CO and produce phenyl radical (3–7) (cf. Figure 5). The last possible option is that the ethyl formate radical results in volatile gases CO_2_ and CO (2–5 or 2–6) (cf. Figure 5). The bond dissociation energy of Reaction 2 is higher than for Reactions 1 and 3 (cf. Figure 5) [22]. Therefore, it is less likely to happen. Furthermore, the formation of CO_2_ in vacuum conditions is lower than CO_2_ formed in the air, which reflects the contribution of oxidation in CO_2_ production [23]. 

The non-radical process could occur as well via intermolecular abstraction of hydrogen through a six-membered intermediate ring. This is known as Norrish Type II (see Figure 5) and leads to carboxylic acid and alkene groups. Carboxylic acid groups could form from both Norrish Type I (1 + 8) and Norrish Type II Reactions (cf. Figure 5). Some researchers proved the dominancy of Norrish Type II over Type I for the formation of carboxylic acid (e.g., [24,25,26]), while others (e.g., [27,28,29]) believe that Norrish Type I is dominating.

#### 2.2.3. Humidity

At ambient temperature, PET is not sensitive to hydrolytic degradation. Above the glass transition temperature (T_g_~67–80 °C), ester bonds are prone to water attack. The ester linkage in PET is hydrolytically cleaved [12], as depicted in Figure 6. Chain scission generates one carboxyl and one hydroxyl end group in exchange for the consumption of one water molecule. There is no chance of crosslinking in hydrolytic degradation.

Above T_g_, the energy of chains increases and chains are more flexible and mobile. This promotes water diffusion. By increasing crystallinity, the water content decreases [14], but it does not mean that crystalline regions are immune to water seepage. However, with increasing fraction of amorphous regions in semicrystalline PET, the water content will increase [6,30]. By continuing, hydrolytic cleavage arising from water diffusion into the amorphous regions, small chains realign and increase the crystallinity (chemi-crystallization phenomenon) [31,32]. Hydrolytic degradation of PET is an autocatalytic reaction, catalyzed by the resulting carboxyl end groups [32]. The hydrolysis rate depends on the intensity of acidic or basic conditions [30]. The rate of hydrolysis increases significantly in both conditions.

#### 2.2.4. Temperature

During manufacturing and processing of PET into fibers or films, a high temperature is needed to melt it. Thermal degradation is a random scission of ester linkage resulting in formation of a vinyl ester and carboxyl end groups [33]. This high temperature in combination with O_2_ causes thermal oxidation and the formation of hydroperoxides, at the methylene group of PET (cf. Section 2.2.1). Hydroperoxides are thermally unstable which accelerates chain scission [34,35,36] and yield mainly new carboxyl, vinylester, and hydroxyl end groups. However, it should be mentioned that the second process generates less vinylester compared to only thermal degradation [37].

The glass transition of PET is between 67 and 80 °C while its melting temperature ranges between 250 and 260 °C [37]. In this way, the producer guarantees that the roofing membrane can be used in temperatures ranging from −40 to +70 °C [38].

### 2.3. Effects of Exposure on PET Degradation 

PET can undergo two forms of degradations: (1) “visible”, in the form of discoloration or cracks; and (2) “invisible”, i.e., changes inside the material that become apparent through different mechanisms only upon imposing additional stress such as loading. Ultimately, the “invisible” changes may also become visible, e.g., by stress-induced cracks.

#### 2.3.1. Discoloration of PET 

It was found by Gok et al. [39] that yellowing is caused by photolytic degradation and haze formation is induced by hydrolytic degradation. When the impacts of light and moisture are coupled, the effects are more serious. Additionally, as time passes, pigment concentration will decrease, more molecules will dissociate, and color fading will happen [40]. Based on [15], vinylester (produced by thermal or thermooxidative degradation) as crosslinkers can form highly conjugated species. Random thermal cleavage of the chains causes the formation of colored species (yellow or brown) (see Figure 7).

#### 2.3.2. Tensile Strength Degradation

In yarns made from PET as semicrystalline plastic, the behavior under tensile forces can be explained by four stages. In Stage 1, reversible elongation of amorphous tie occurs (widening angle of zigzag chains restricted to linear elasticity). This arises because of the elongation of the molecular chains by bond stretching and orientation in the direction of applied stress. Then in Stages 2 and 3, as the strain keeps increasing, chains cannot stretch anymore. Crystalline regions slip past each other (Stage 2) and also might separate to crystalline segments (Stage 3). The tie chains of amorphous structures attach these segments to each other. This is the start of irreversible plastic deformation. Finally, in Stage 4, these crystalline segments and tie chains of amorphous region stretch along with tensile forces. When the permanent plastic deformation starts, the cross-section area begins to decrease (necking). In the vicinity of the necked region, crystalline parts deform extensively while amorphous regions stretch in the direction of the applied stress. When these extended chains cannot resist further deformation, the necked regions will deepen continuously leading to local failure [14].

Al-Azzawi [14] observed unchanged strain amounts during an early state of outdoor weathering. She used this observation to explain that weathering at an early stage occurs mostly in amorphous regions and has not any pronounced effect on crystallinity. It should be noted that, by increasing the number of chain scissions, chains are getting shorter and cannot elongate as in their original state. Al-Azzawi reported a sharp decrease of the stress and strain at failure at an early stage of artificial weathering (exposure to 1000 h). It was assumed by Al-Azzawi [14] that this behavior is caused by the rapid growth of inevitable microcracks due to artificial weathering. However, she observed that 1000 h of exposure for outdoor specimens was not enough to achieve a growth of these microcracks. For those specimens, Al-Azzawi observed that stress and strain at failure did not change after 1000 h of natural weathering compared to the virgin material.

#### 2.3.3. Photodegradation

Formation of both carboxyl [27,28,29] and/or carbonyl end groups [41] are accompanied with mechanical properties deterioration such as tensile strength and elongation at break. Oxidation also increases the chance of crosslinking. One procedure of photodegradation is chain scission with the following effects: shorter chains do not have enough length to form crystalline segments, therefore, crystallinity decreases, and shorter chains have less mass, so the amount of absorbed energy to relax is lower (transition from glassy to rubbery state, T_g_). This means earlier relaxation of the chains at lower temperatures [12]. Finally, during chain scission, shorter chains with more ends per unit volume are created [42], which results in higher free volumes and formation of microcracks [13,14]. Higher free volumes are more permeable to water and oxygen, which eventually trigger microcracks [13]. These microcracks extend with the progress of weathering mechanisms. For example, in long-term weathering, moisture, which is trapped in the crack tips, could freeze and expand during winter causing extra tension and crack growth. The chain scission process is faster in amorphous regions due to more free volumes between the chains which facilitate oxygen diffusion and water penetration. During the tension test, microcracks at degraded layers develop to undegraded inside layers. This makes specimens more susceptible to failure under lower stresses [14]. It should be considered that microcracks must reach a critical crack length to initiate earlier failures. Moreover, chain scission and crosslinking transfer linear macromolecules with low crosslink density to dendritic large crosslink density ones. This makes PET brittle to develop cracks [6]. The presence of microcracks increases the roughness of surfaces as well [14]. Philip and Al-Azzawi [13] observed that surface degradation (discoloration and microcracks) occurs at the early stage of UV exposure.

#### 2.3.4. Hydrolytic Degradation

For architectural structures on rare occasions, the service temperature might rise beyond the glass transition temperature of PET; therefore, the possibility of hydrolysis is very low [43]. Overall, at atmospheric temperature, neutral water cannot change PET yarns, but it transports some risky substances which are prone to the growth of fungi or bacteria [8]. Acid rain and water in contact with concrete are examples of acidic and basic attacks to membrane PET. Ducoulombier et al. [44] observed that, when the temperature is lower than T_g_, hydrolysis in the basic medium results in quick deterioration of the PET properties. They assumed that an attack of OH- causes an erosion of the PET fabric surface coupled with weight loss. 

#### 2.3.5. Thermal Degradation

The initial step of thermal degradation is the chain scission of the ester linkage, which results in molecular weight decrease [32]. Increasing temperature is known to facilitate ageing; nevertheless, during artificial ageing, it must be limited to T_g_, because the structure of the fabric should not be altered by the sole action of a temperature higher than the service temperature, as, in such a case, aging would no longer reflect the reality [44].

## 3. Materials and Methods

### 3.1. General

The presented investigation focused on weathering the substrate of PET-PVC, a woven PET fabric (see Figure 8). As outlined in Section 2.1, the coating might be removed, e.g., by local damages, during the service life of the membrane structure. Hence, it might be useful to scrutinize the weathering mechanisms of the load-bearing woven PET fabrics.

### 3.2. Materials

Within the presented study, woven PET fabrics were investigated, which are the base material for PET-PVC Type II and III fabrics (see the specifications in Table 1). In the context of this paper, the analyzed type of polyester is precisely called polyethylene terephthalate (PET). The type classification is applied for the coated fabrics and was based on ref. [45]. Types II and III were selected because they are very frequently used in outdoor architectural applications and thus reflect a huge share of the market. The PET fabrics were taken from the production before coating and were also termed Types II and III in this study to associate them to the respective coated materials. The higher strength of Type III was realized by a higher yarn density (see Table 1). Both uncoated fabrics were not pigmented and white. It should be noted that material producer is Mehler Texnologies GmbH, Hückelhoven, Germany.

### 3.3. Methods

#### 3.3.1. Artificial Weathering

Four different artificial weathering techniques were carried out to simulate impacts of humidity, temperature, and UV light. The methods are listed in Table 2. In all methods, a QUV accelerated weathering tester was utilized with UV-A lamps (emission peak at 340 nm and irradiance of 0.76 W/m^2^, comparable with global summer noon, normal incidence on the summer solstice [46]). Methods M1–M3 were used to investigate the impact of temperature (up to T = 80 °C, i.e., not higher than T_g_ of PET), humidity, and post photodegradation effects (dark cycles). Method M4 was based on EN ISO 4892-3 (2016, Part 3, Method 1) [47]. It simulates outdoor weathering (morning dew followed by sunlight) by alternating sequences of UV light, heat, and condensing humidity [39]. As a standardized weathering cycle, it was used to survey the impact of the exposure time. It should be noted that all cycles were constantly running, i.e., without any break.

#### 3.3.2. Uniaxial Tensile Tests

Virgin and weathered materials were tested in a uniaxial constant rate extension (CRE) machine according to EN ISO 13934-1 [48]. For tensile strength and breaking strain determination, two sets of test specimens consisting of three samples each were tested, one in warp and the other in weft direction. It should be noted that, for uncoated fabrics, yarns are not woven together firmly and can be separated from each other easily. For this reason, according to EN ISO 13934-1, two long edges of specimens should be free of dominating yarns about 10 mm or 15 yarns (longest yarns) by pulling out. In this way, the middle yarns inside the specimens are fix enough and they might not move during testing. The width of the samples used for the uncoated fabrics (90 mm) was bigger than the standard size (50 mm). 

#### 3.3.3. Intrinsic Viscosity and Molecular Weight Measurement

The inherent viscosity of PET was measured according to ASTM D4603-03 [49], with 0.5% PET in a mixture phenol: 1,1,2,2-tetrachloroethane (60:40) as solvent. A glass capillary viscometer (Ubbelohde) was used at 30 °C. Relative and inherent viscosity (η_*r*_ and η_inh_) are defined as:(1)$$\eta_{r}=\frac{t}{t_{0}}$$
with *t* as average solution flow time and *t*_0_ as average solvent flow time, and
(2)$$\eta_{\mathrm{inh}}=\frac{\ln\eta_{r}}{C}$$
with *C* as the concentration of the PET solution (g/dL).

Finally, the intrinsic viscosity was calculated using the Billmeyer relationship [48,50]:(3)$$\eta =0.25\frac{\left(\eta_{r}-1+3\ln\eta_{r}\right)}{C}$$

Allen et al. [51] determined empirically the relationship between intrinsic viscosity and number average molecular weight (*M_n_*) or weight average molecular weight (*M_w_*) as follows:(4)$$\eta =1.7\times 10^{-4}{\left[\bar{M_{n}}\right]}^{0.83}=1.6\times 10^{-4}{\left[\bar{M_{w}}\right]}^{0.76}$$

The number of chain scissions per molecule [51] can be calculated by:(5)$$\mathrm{Chain}\;\mathrm{scissions}\;\mathrm{per}\;\mathrm{molecule}=\;\frac{\bar{M_{n0}}}{\bar{M_{nt}}}-1$$
where *M*_*n*0_ and *M_nt_* are the number average molecular weights at time zero and *t*, respectively. 

#### 3.3.4. Infrared (IR) Spectroscopy

An IR Tracer-100 spectrometer from Shimadzu (Kyoto, Japan) with a diamond attenuated total reflectance (ATR) accessory (MIRacle 10 with a ZnSe-diamond) was used to record IR spectra of the pristine and degraded yarns. The spectra were taken between 400 and 4000 cm^−1^ with 20 scans and resolution of 4. Happ-Genzel was utilized as the apodization function. Because of differences in the specimen properties such as dimensions or different contact between sample and ATR crystal, the measured absolute values were normalized against an absorbance band (1404 cm^−1^, belonging to C-C stretching vibration of aromatic skeleton ring) which does not change during weathering degradation. Two sets of test specimens consisting of three samples each were tested, one in warp and the other in weft direction (by fraying warp/weft yarns of specimens). Characteristic absorptions of PET that can potentially be used as mechanistic variables in data analysis are presented in the Appendix A. 

#### 3.3.5. Optical Microscopy

Micrographs were taken using a Keyence digital microscope VHX-70 (Osaka, Japan). 

#### 3.3.6. Exploratory Data Analysis

The approach including stressor, system-level mechanistic response, and performance level response was utilized to develop a stepwise regression model that ties stressors (conditions that affect degradation) and responses [12]. In this degradation pathway model, exposure time and tensile strength were considered as the main stressor (independent variable) and performance level response, respectively. Mechanistic variables were selected from viscosity and IR evaluations. The best fit between two variables was chosen according to the Pearson correlation with the help of Statistical Package for the Social Sciences (SPSS). Only in the case that the significance level (a number between 0 and 1 representing the probability that the data would have arisen if the null hypothesis, no relationship between pairs of data, were true) was less than 0.05, a meaningful correlation between parameters existed. The Pearson correlation only reports a univariant relationship. For developing the mechanistic pathway equations, mechanistic variable equations were substituted into the equation for the complete pathway [18].

#### 3.3.7. Yellowness Index Measurement

Yellowness index (YI) measurement was conducted using Lambda 950-S UV/Vis/NIR Spectromter (PerkinElmer LAS GmbH, Rodgau, Germany) with Lambda 950 InGaAs Sphere. YI is the measure of yellowing of a sample and calculated using transmission spectrum in the UV–Vis region (in the reflectance mode) as defined by DIN 6167 [51]. YI was measured at Deutsches Textilforschungszentrum Nord-West gGmbH, Krefeld, Germany.

## 4. Results

### 4.1. Impact of Temperature, Humidity, and Post Photodegradation Period

Uniaxial tensile tests were used to study the influence of the main weathering factors. Mean stress–strain curves obtained for PET fabrics (from 3 specimens) after exposure according to Methods M1–M3 are almost the same. The data in Figure 9a exemplify this for 50 °C. Figure 9b depicts results for Methods M1-1–M1-4, i.e., for different temperatures. Figure 10 presents the resulting tensile strength as a function of temperature for weathering according to the three methods. Again, no significant difference can be seen. This can be interpreted as low probability of occurrence of hydrolysis in the temperature range below glass transition and negligible influence of post photodegradation in dark cycles. This means the non-existing water impact, as described in Section 2.3.4, is confirmed for the PET fabric investigated here. Regarding post photodegradation, there is no difference between the stress–strain curves resulting from Method M2 (including dark cycles) compared to the other methods (cf. Figure 9a). This clarifies that degradation does not continue after the UV sources are switched off. Hence, the only influential impact on degradation is a combination of UV light and heat.

Figure 10 provides evidence that the average tensile strength does not change due to the different used weathering techniques while the temperature is the same. However, with increasing temperature, the tensile strength deteriorates more quickly, which means that heat facilitates photodegradation (see Figure 9b and Figure 10).

### 4.2. Impact of Weathering Time

A standard test method that simulates outdoor weathering (morning dew followed by sunlight), at high temperatures (50 and 60 °C) to accelerate the effects, was used to analyze the influence of the time on the tensile strength as the key property for the membrane performance as well as on structural changes of PET. 

#### 4.2.1. Tensile Strength

Figure 11 reveals that the general trend of the stress–strain curves in both warp and weft directions, including three specific regions of woven fabric stress–strain paths (inter-fiber friction, decrimping, and yarn extension [1]), remains unchanged up to 480 h of weathering. After 480 h, the decrimping region disappears, i.e., filaments break before the yarns decrimp.

Residual mean values of the tensile strength given from three weathered specimens are depicted in Figure 12. According to this figure, a strong decreasing trend is governing the charts; it shows that, with the passage of time, weathering causes negative changes in tensile strength. This decreasing trend was already observed by Philip and Al-Azzawi [13]. For developing a better mechanistic understanding of the phenomena, different properties (responses) such as molecular weight and number of chain scissions were examined. 

#### 4.2.2. Molecular Weight

Figure 13 exhibits a decrease in the molecular weight and, consequently, an increase in the number of chain scissions with the passage of time. This is in accordance with results of other investigations [25,26,28,29]; for photodegradation of virgin PET, a decrease of the molecular weight and deterioration of the mechanical properties (e.g., tensile strength and elongation at break) were observed, which indicate polymer chain scission as governing mechanism. Except for two weathering time periods (382 and 480 h), the molecular weight of the yarn used in warp direction is always higher than that in weft direction; the number of chain scission is always lower in warp direction (cf. Figure 13). This shows that weft yarns are more sensitive to weathering effects, which is consistent with findings reported in the literature [6,52] for PET-PVC fabrics. This may arise from either the weaving pattern or the weaving process, i.e., pre-stressed warp and slack weft yarns.

The rate of changes in molecular weight and the number of chain scissions, as well as that for tensile strength, is always higher at an early stage of weathering (compare Figure 12 and Figure 14). The effects reach a plateau after approximately 1000 h. This implies possibly the barrier effect of by-products that has been observed in another work [13]. Random chain scission takes place in two steps: a very rapid initial step, followed by a normal step [53]. The correlation between the tensile strength and the molecular weight is presented in Figure 14. The S-shape curves are in accordance with those in [54] and indicate a relationship with three definite regimes. The regimes are defined based on the occurrence of the breaking point of the specimens at three definite regions of the stress–strain curve (see Figure 11). Breaking happens for Regimes I–III at inter-fiber, decrimping, and yarn-extension regions, respectively.

#### 4.2.3. IR Spectroscopy

Figure 15 illustrates the FTIR spectra of PET Type II material, in virgin and weathered states. 

Figure 16a depicts increasing trends in C=O of carboxylic acid, as a signature of chain scission in both warp and weft directions [55]. The ratio of bands that are attributed to trans and gauche conformations of the (CH_2_) wagging at 1343 and 1376 cm^−1^ can be assumed as crystallinity index during degradation, since trans form is favored by crystalline phase [56]. According to Figure 16b, this ratio decreases significantly for weft yarns of PET by the weathering time, which is already proven in [57]. Crystallinity decrease means more permeability to water and oxygen diffusion (cf. Section 2.3.4). The crystallinity can be measured by the degree of crystallinity as 100×A973/A1018 [56], where A973 and A1018 are the absorption rate of asymmetric stretching vibration of C–O of ethylene glycol in transform and in plane ring deformation (C–H) in the amorphous phase, respectively [26]. The degree of crystallinity decreases for warp yarns of PET by the weathering time. The process of chain scission produces shorter chains, which are not enough to form crystalline segments, so the crystallinity decreased [14].

#### 4.2.4. Discoloration

Figure 17 shows the gradual discoloration of the PET specimens by increasing weathering duration. Relevant photodegradation-induced discoloration is discussed in Section 2.3.1. 

Figure 18 shows the change of the yellowness index (YI) in all exposure times. By augmentation of the exposure time, the YI increases. At the early stages of weathering, the growth rate of YI is not as remarkable as for more prolonged time periods (from 1440 to 4684 h). The findings of Gok et al. [39] prove the increase of YI with the passage of time, when they are exposed to artificial UVA lights (1.55 W/m^2^). Apparently, the discoloration cannot be a good indicator of the tensile strength decrease. While the tensile strength was observed to decrease rapidly for the uncoated PET fabric (cf. Figure 12), significant discoloration occurred only in later stages.

### 4.3. Mechanism of Tensile Strength Degradation with Weathering Time 

In this section, pathway diagrams for changes under artificial accelerated weathering condition (M4) are presented for the tensile strength in order to clarify its complex degradation mechanisms. In these diagrams, time is the main stressor, and the tensile strength is the performance level response variable. Potential mechanistic variables are all measured values presented and discussed in Section 4.2. In the pathway modeling with the implication of the Pearson correlation, variables are not considered relevant when the significance level is equal to or higher than 0.05. Figure 19 shows the obtained pathway diagrams. The Pearson’s correlation coefficient (r) is a measure of the strength of the association between the two variables, which ranges for continuous (interval level) data from -1 (negative linear correlation) to +1 (positive linear correlation). Equations used for describing the tensile strength-weathering time period curves in Figure 20 were developed by using the stepwise regression model (see Appendix A). In this way, the best fitting polynomial functions were selected. Here, indirect pathway equations were obtained by substituting mechanistic variables equations into the complete path [12]. Additionally, M_n_, M_w_, and number of chain scissions are related to each other (cf. Section 3.3.3). They behave in the same way and only two of them (M_n_ and chain scission) were included in the pathway equations.

Plots of pairwise relationship pathway diagrams between stressors and responses are shown in Figure 20. In warp direction, the mechanistic pathway through M_n_ (t → M_n_ → Ts) or chain scission depicts (t → chain scission → Ts) almost perfectly match the direct path (t → Ts), especially in the medium exposure time periods. The mechanistic pathway through C=O of the COOH group has an acceptable match between 200 and 1000 h. For weft direction, the best but not perfect match belongs to t → M_n_ or chain scission → Ts. C=O of COOH and the crystallinity variables underestimate the tensile strength at very early and late exposure stages while overestimating the tensile strength in the mid-exposure stage. In general, the relationship t → M_n_ → Ts or t → chain scission→ Ts can act as an alternative reasonable path for the tensile strength degradation by time, specifically in warp direction.

### 4.4. Comparison of Weathering Deterioration of PET Woven Fabric Type II and III

Tensile strength changes of two types of investigated uncoated PET woven fabrics, Type II and III, during about 120 h of artificial weathering by Method M4 are shown in Figure 21. The change of the tensile strength due to weathering is about 45.9% (in warp) and 46.2% (in weft) for Type II and 40.2% (in warp) and 42.6% (in weft) for Type III materials. Considering the experimental error, no significant differences between warp and weft directions are observed. These results illustrate a slightly higher degree of weathering sensitivity for uncoated PET fabrics Type II than for uncoated PET fabrics Type III.

## 5. Conclusions

In this contribution, different weathering mechanisms, which influence directly or indirectly the tensile strength of uncoated woven PET fabrics, were surveyed experimentally. Different accelerated weathering exposures were applied and the changes in the tensile strength were evaluated. Concurrently, changes in the polymer structures were evaluated by IR spectroscopy and viscosity measurements (revealing information about molecular weight and chain scission). Measured responses were used to generate pathway diagrams. These pathway models were successfully established by the contributions from mechanistic variables, such as the molecular weight changes providing the most obvious correlations. Taking advantage of all presented results, the following conclusions can be obtained:(1)In artificial weathering, it was found that water induced cycles at temperatures below T_g_ do not have any impact on the tensile strength degradation of the investigated PET fabric.(2)Increasing the temperature could accelerate artificial weathering (but it should anyway be limited to values below T_g_).(3)Alternating dark and light cycles (for mimicking day and night) and possible post photodegradation mechanisms were not critical for the tensile strength of uncoated PET material.(4)The most influential mechanism on the tensile strength degradation under combined heat, humidity, and UV attacks was the chain scission caused by photodegradation.(5)By increasing the exposure time periods, both tensile strength and molecular weight decreased. Based on the similarity between the direct stress/performance level response pathway (t → Ts) and the multistep pathway, via mechanistic level response (t → M_n_ or chain scission → Ts), it can be inferred that changes of the tensile strength and molecular weight directly correlate with each other. For the investigated woven PET fabric, a very good correlation was found for the fabric’s warp direction, whereas the correlation in weft direction can still be improved. It is assumed that differences in the warp and weft yarns densities and/or between pre-stressed warp (inserted during weaving) and slack weft yarns cause such a difference.(6)The investigated woven PET fabric Type II is a little more sensitive than Type III to tensile strength deterioration under weathering effects.

Overall, uncoated PET woven fabrics as part of PET-PVC fabrics for architectural membranes resulting from coating removal due to local damages such as abrasion or coating pulverization have no significant resistance to UV and high-temperature influences arising from outdoor weathering.

## Figures and Tables

**Figure 1 materials-14-00618-f001:**
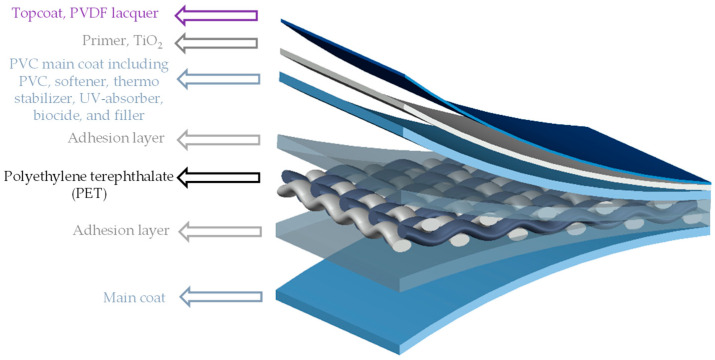
Different layers of PVC-coated PET woven fabric (PET-PVC), schematic view.

**Figure 2 materials-14-00618-f002:**
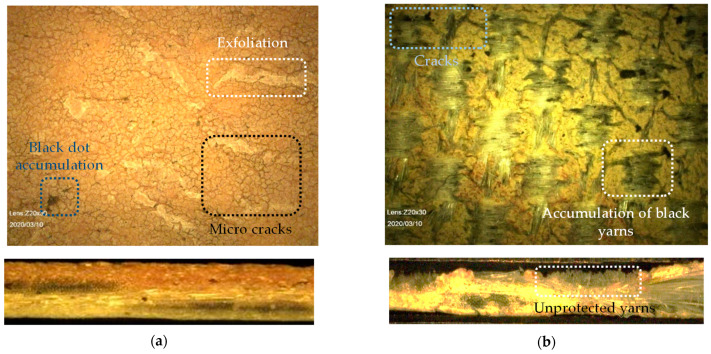
Dismantled fabric taken from realized projects: (**a**) 23-year-old PET-PVC Type III; and (**b**) 38-year-old PET-PVC Type II with PVDF top coating, taken by digital microscope (see Section 3.3.5).

**Figure 3 materials-14-00618-f003:**
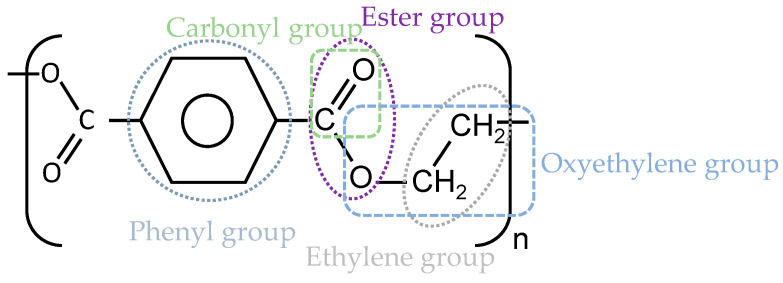
The repeating unit of PET (C_10_H_8_O_4_).

**Figure 4 materials-14-00618-f004:**
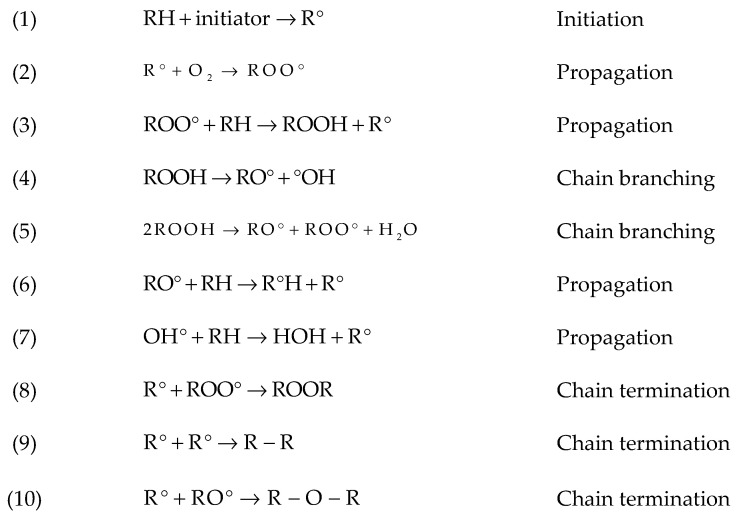
Overview on hydrocarbon oxidation reactions [16]. RH, polymer; ROO°, peroxy radical; ROOH, hydroperoxide; RO°, alkoxyl radical; °OH, hydroxyl radical.

**Figure 5 materials-14-00618-f005:**
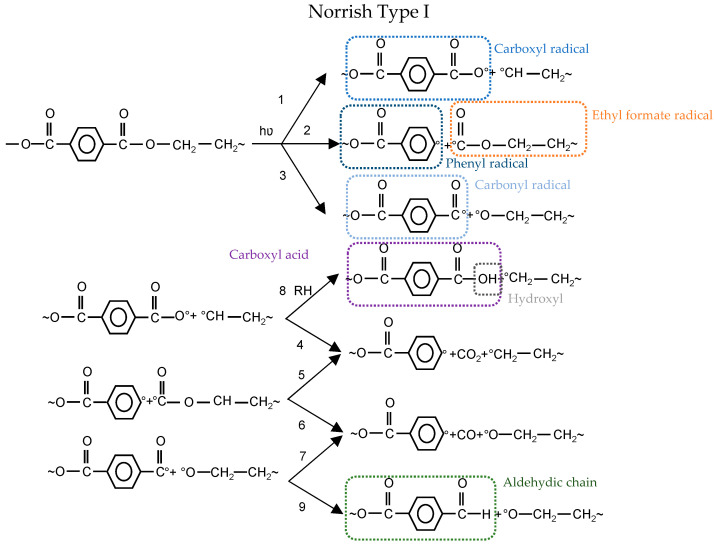
Photochemical degradation mechanisms: (**top**) Norrish Type I; and (**bottom**) Norrish Type II [22].

**Figure 6 materials-14-00618-f006:**
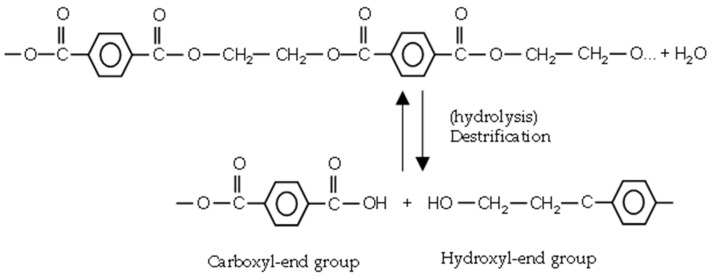
Hydrolysis reaction of PET.

**Figure 7 materials-14-00618-f007:**
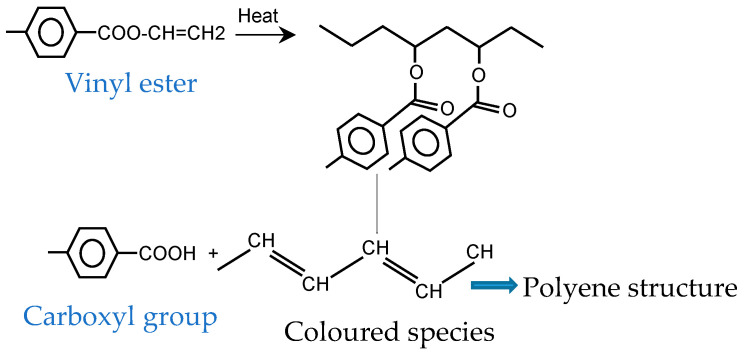
Formation of colored species in PET-derived structures.

**Figure 8 materials-14-00618-f008:**
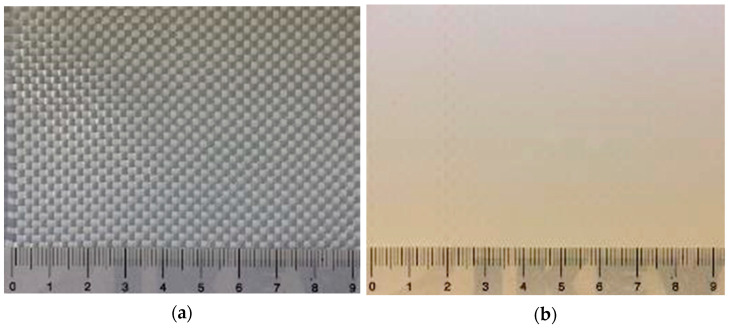
PET-PVC Type III: (**a**) uncoated woven PET fabric; and (**b**) PVC coated woven PET fabric.

**Figure 9 materials-14-00618-f009:**
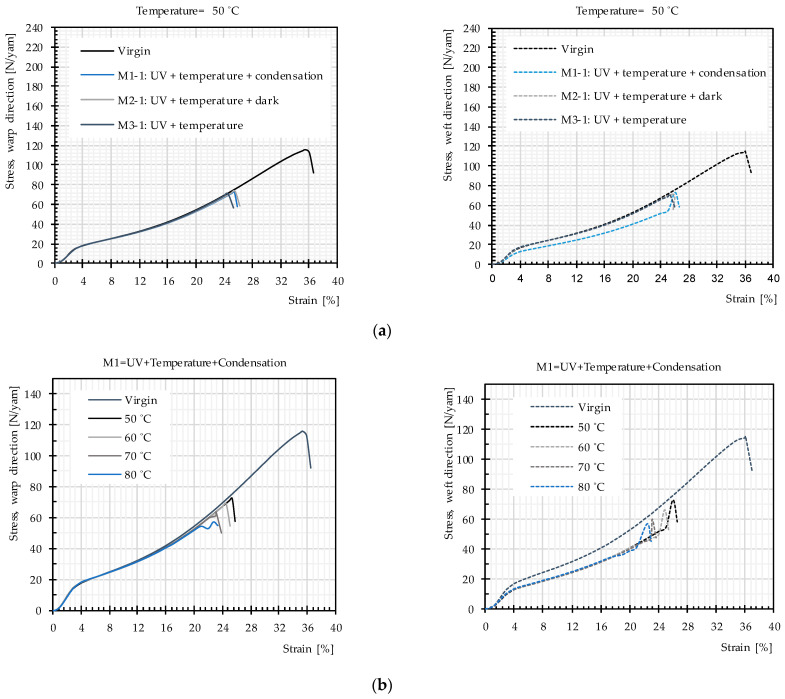
Mean stress–strain curves, uncoated PET Type III: (**a**) artificial weathering methods (M1-1–M3-1); and (**b**) artificial weathering Method M1.

**Figure 10 materials-14-00618-f010:**
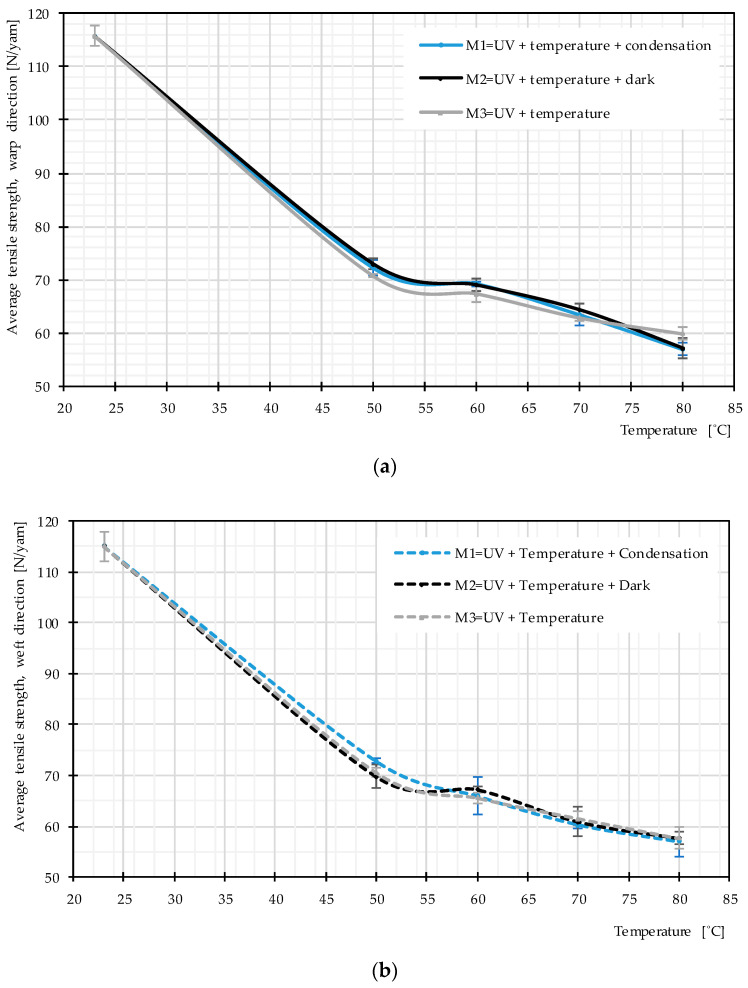
Average tensile strength, various weathering techniques at different temperature, uncoated PET Type III: (**a**) warp direction; and (**b**) weft direction.

**Figure 11 materials-14-00618-f011:**
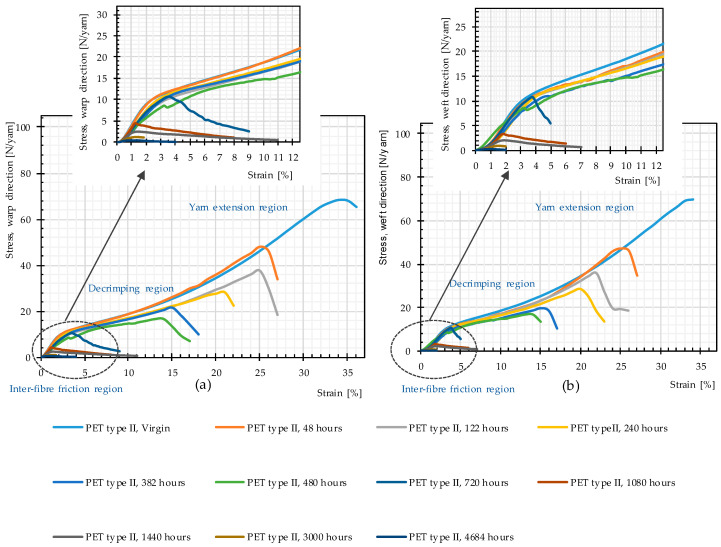
Stress–strain curves, weathering Method M4, different weathering durations, uncoated PET Type II: (**a**) warp direction; and (**b**) weft direction.

**Figure 12 materials-14-00618-f012:**
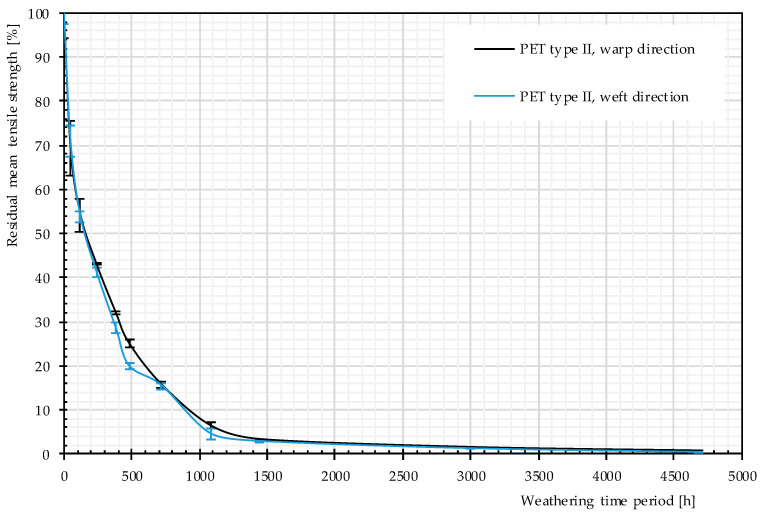
Residual strength, different artificial weathering durations (Method M4), PET Type II.

**Figure 13 materials-14-00618-f013:**
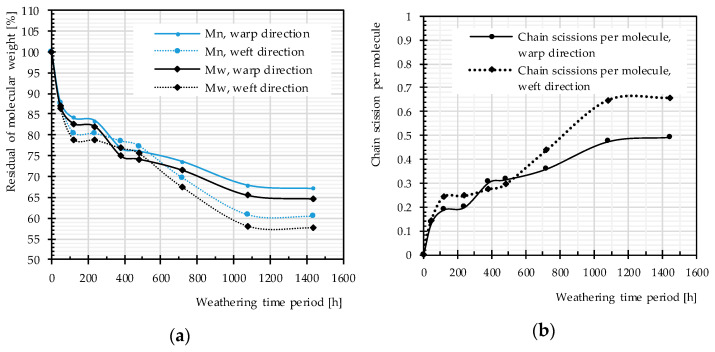
Different artificial weathering durations (Method M4), uncoated PET Type II: (**a**) molecular weight changes; and (**b**) chain scission changes.

**Figure 14 materials-14-00618-f014:**
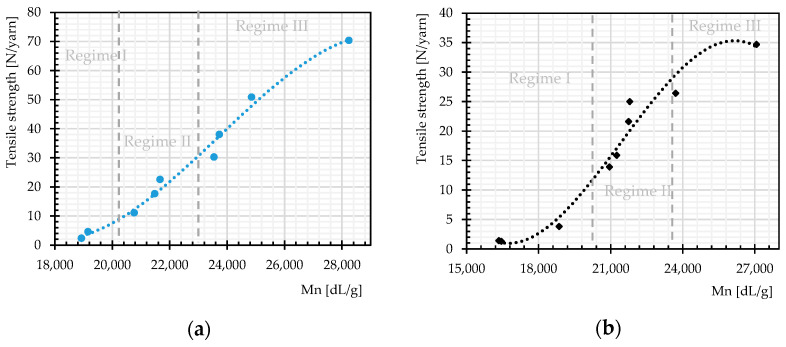
Tensile strength as function of molecular weight regimes, different weathering durations: (**a**) warp direction; and (**b**) weft direction.

**Figure 15 materials-14-00618-f015:**
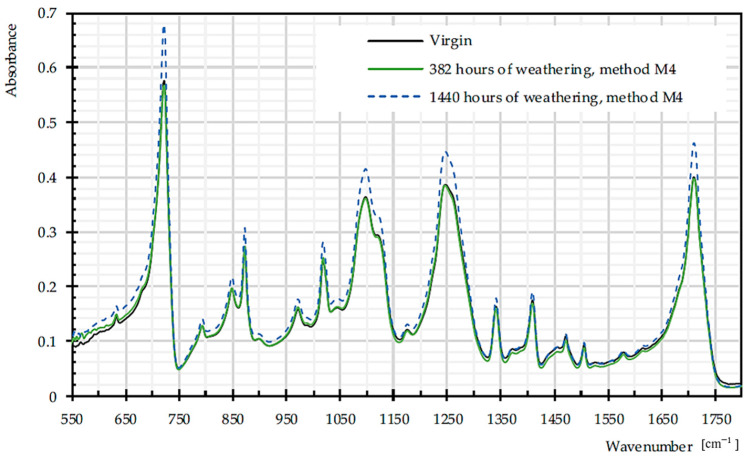
Mean FTIR spectra for virgin and dismantled PET Type II, given from three specimens.

**Figure 16 materials-14-00618-f016:**
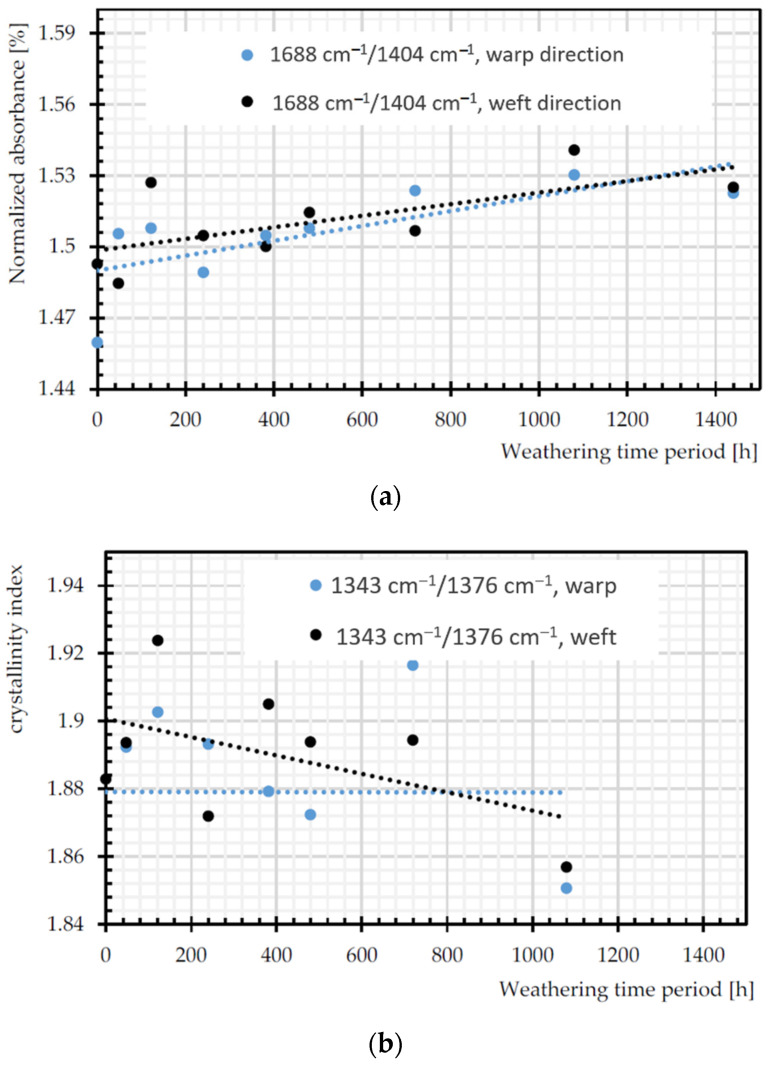
Normalized absorbance changes of: (**a**) C=O of carboxylic acid (1688 cm^−1^); (**b**) crystallinity index (A1343 cm^−1^/A1376 cm^−1^); and (**c**) degree of crystallinity (100 × (A973 cm^−1^/A1018 cm^−1^)).

**Figure 17 materials-14-00618-f017:**
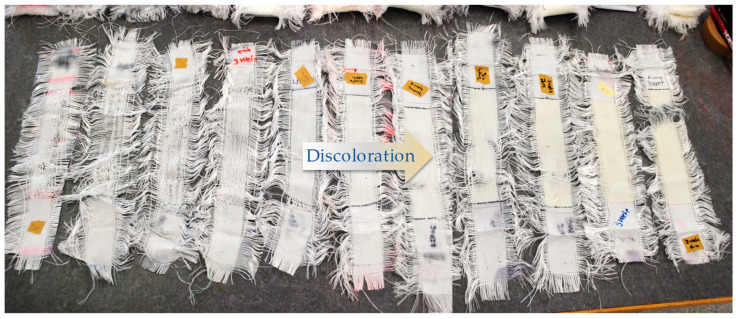
Discoloration, left to right: from 0 to 4684 h, uncoated PET Type II.

**Figure 18 materials-14-00618-f018:**
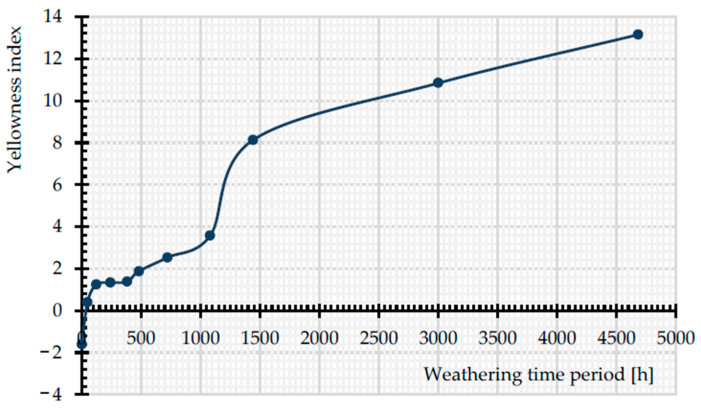
Change in YI with exposure time, from 0 to 4684 h, uncoated PET Type II.

**Figure 19 materials-14-00618-f019:**
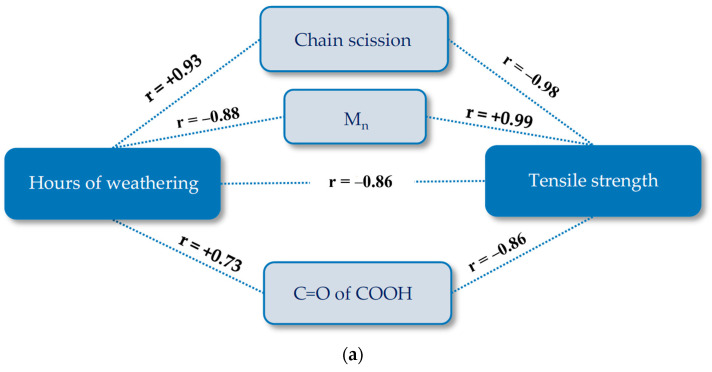
Pathway diagram for the tensile strength deterioration of uncoated PET Type II under artificial exposure (Method M4): (**a**) warp direction; and (**b**) weft direction. R, Pearson’s correlation coefficient; C=O of COOH of carboxylic acid (1688 cm^−1^); crystallinity (1343 cm^−1^/1376 cm^−1^).

**Figure 20 materials-14-00618-f020:**
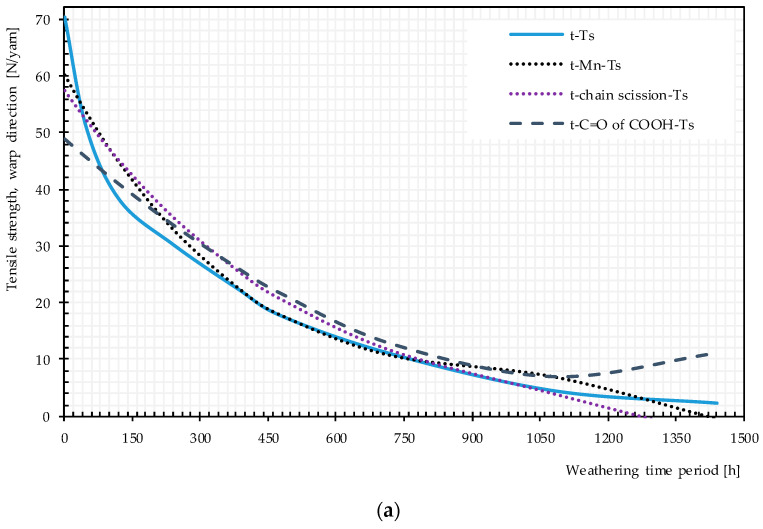
Direct and multistep pathway diagrams for the tensile strength deterioration of uncoated PET Type II under artificial exposure Method M4: (**a**) warp direction; and (**b**) weft direction. t, time of exposure (hour); Ts, tensile strength.

**Figure 21 materials-14-00618-f021:**
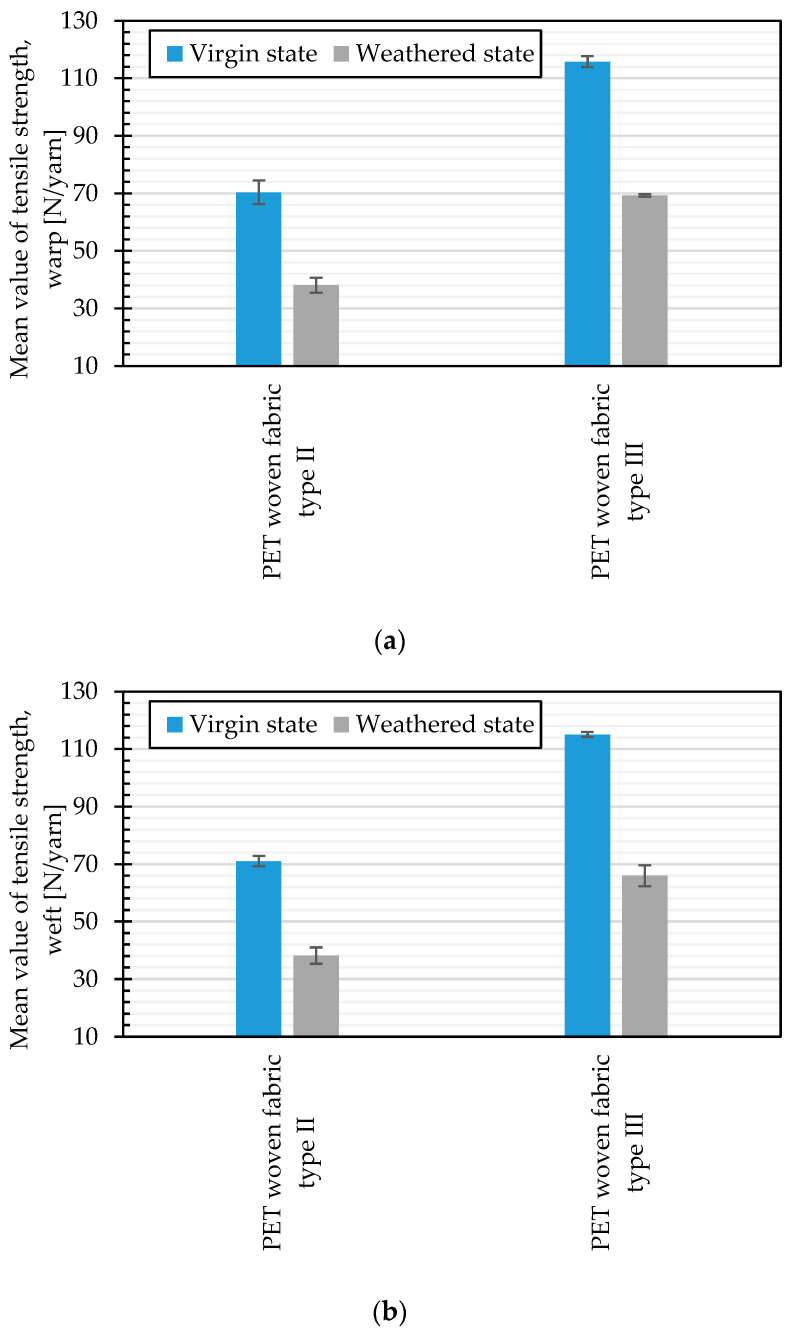
Comparison of the mean tensile strength, virgin, and weathered states (artificial weathering, Method M4, 120 h): (**a**) warp direction; and (**b**) weft direction.

**Table 1 materials-14-00618-t001:** Specifications of woven PET materials.

Material	Weave Pattern	Yarn	Mass (g/m^2^)	Yarn Count (cm^−1^) ^1^, Warp/Weft	Yarn Density (dtex) ^2^	Thickness (mm)
PET Type II	Panama 2/2	PET low wick	337	11/12.5	1100	0.348
PET Type III			363.1	10.8/10.5	1670	0.440

^1^ Mass per unit length (0.1 g/km), taken from datasheets of coated fabrics (the same producer). ^2^ Taken from datasheets of coated fabrics (the same producer) since, in uncoated woven PET fabrics, yarns are not woven firmly and could move easily.

**Table 2 materials-14-00618-t002:** Artificial weathering cycles.

Method	Standard	Cycles	Duration
M1-1	-	10 × (8 h radiation at 50 °C ± 3 K and 4 h condensation at 50 °C ± 3 K)	Each: 80 h radiation + 40 h condensation = 120 h
M1-2	-	10 × (8 h radiation at 60 °C ± 3 K and 4 h condensation at 50 °C ± 3 K)	
M1-3	-	10 × (8 h radiation at 70 °C ± 3 K and 4 h condensation at 50 °C ± 3 K)	
M1-4	-	10 × (8 h radiation at 80 °C ± 3 K and 4 h condensation at 50 °C ± 3 K)	
M2-1	-	10 × (8 h radiation at 50 °C ± 3 K and 4 h dark at 50 °C ± 3 K)	Each: 80 h radiation + 40 h dark = 120 h
M2-2	-	10 × (8 h radiation at 60 °C ± 3 K and 4 h dark at 50 °C ± 3 K)	
M2-3	-	10 × (8 h radiation at 70 °C ± 3 K and 4 h dark at 50 °C ± 3 K)	
M2-4	-	10 × (8 h radiation at 80 °C ± 3 K and 4 h dark at 50 °C ± 3 K)	
M3-1	-	10 × (8 h radiation at 50 °C ± 3 K)	Each: 80 h radiation
M3-2	-	10 × (8 h radiation at 60 °C ± 3 K)	
M3-3	-	10 × (8 h radiation at 70 °C ± 3 K)	
M3-4	-	10 × (8 h radiation at 80 °C ± 3 K)	
M4	EN ISO 4892-3:2016	8 h radiation at 60 °C ± 3 K and 4 h condensation at 50 °C ± 3 K	48, 122, 240, 382, 480, 720, 1080, 1440, 3000, 4684 h

## Data Availability

Data is contained within the article or Appendix A.

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
