# Peer review of "Artificial Weathering Mechanisms of Uncoated Structural Polyethylene Terephthalate Fabrics with Focus on Tensile Strength Degradation"

_materials, 2021, doi:10.3390/ma14030618_

Round 1

Reviewer 1 Report

Review Materials-1037637

Even thou there are instruments for fabric weathering e.g. Xenotest (SDl Atlas) which corresponds to  ISO 4892-2, 11341, 105-B04, 105-B10, this paper is very interesting using different chambers for different environment. There achieved results are well explained, however some data is missing to confirm these results.

Page p.2. line 47… In the textile area, polyethylene terephthalate (PET) is named PES,.. not exactly… PES is abbreviation for polyester in general (PET, PLA PBT etc.). The authors are correct that officially PES in material science stands for.. Please rephrase the sentence, or delete. It is not necessary in introduction at all. You can write in material part: “In the context of this paper, the analysed type of polyester is precisely polyethylene terephthalate (PET).”

p.9, l. 283/4  the producer of fabric is missing

RESULTS

In my opinion, from the results of tensile strength in warp direction mechanical damage should be calculated:

                  [%]      

Instead the Figure 17, exact results should be given. The remission should be measured on remission spectrophotometer or similar, and afterwards the results should be given as yellowing index, YI according to DIN 6167. Description of yellowness of near-white or near-colourless materials. Usually the instrument automatically calculate this value and for white fabrics it is the most common representation of results (for finishing or coating) or as color fastnes ISO 105-B02/B04.

Degree of crystallinity using FT-IR ATR results should be calculated. Here are some references how to do so:

  • Tarbuk, Anita; Đorđević, Dragan; Flinčec Grgac, Sandra; Kodrić, Marija; Magovac, Eva; Čorak, Ivana: The influence of lipase surface modification to polyester crystallinity and absorbility, BOOK OF PROCEEDINGS 13th International Scientific Professional Symposium TEXTILE SCIENCE & ECONOMY Zagreb: University of Zagreb Faculty of Textile Technology, 2020. pp. 33-38. http://tzg.ttf.unizg.hr/wp-content/uploads/2020/10/TZG-2020_Book-of-Proceedings.pdf
  • Grime D, Ward IM. The assignment of infra-red absorptions and rotational isomerism in polyethylene terephthalate and related compounds. Trans Faraday Soc. 1958;54(0):959. doi:10.1039/tf9585400959
  • Yazdanian M, Ward IM, Brody H. An infra-red study of the structure of oriented poly(ethylene terephthalate) fibres. Polymer (Guildf). 1985;26(12):1779-1790. doi:10.1016/0032-3861(85)90003-5
  • Ward IM, Wilding MA. Infra-red and Raman spectra of poly(m-methylene terephthalate) polymers. Polymer (Guildf). 1977;18(4):327-335. doi:10.1016/0032-3861(77)90077-5
  • Farrow G, Bagley J. The Measurement of Molecular Orientation in Polyethylene Terephthalate Filaments by X-Ray Diffraction. Text Res J. 1962;32(7):587-598. doi:10.1177/004051756203200709

Reviewer 2 Report

Very interesting manuscript discussing the weathering mechanisms of PET fabrics. Authors discussed the influence of several parameters (e.g. photo-degradation, humidity, temperature) on the visible and invisible degradation process (resulting in discoloration, tensile strength degradation, photodegradation, hydrolytic degradation, and thermal degradation) of the PET fabric. To assess the rate and the range of degradation Authors used several physicochemical and instrumental methods, like tensile tests, determination of the intrinsic viscosity, molecular weight determination, as well as IR spectroscopy and optical microscopy. The samples for the research were prepared by using the artificial weathering. The results are well discussed, taking into account each of the parameter influencing the degradation. Eventually, the possible pathways of the tensile strength deterioration of PET were proposed and discussed. The conclusions are consistent with the findings.The manuscript fits well the profile of the journal. Paper is very well written and can be published in the submitted form. The only comment refers to the line 102, where an error was displayed "Reference source not found".

I would recommend the publication of the manuscript in Materials.

Reviewer 3 Report

The article comprehensively analysis the impact of temperature, humidity, etc. on the tensile strength and discoloration of polyethylene terephtalate fabrics. The article is nicely written, the state of the art is presented quite extensively. However, I have missed the part of discussion of the results with other authors results. Additionally, the information about the materials used (manufacturers, countries of origin, etc.) were missing as well. Moreover, conclusions do not reflect that two types of PET materials were analysed. Which one is better? Please add additional observations and improve the conclusions part. 

Round 2

Reviewer 3 Report

Authors have taken into consideration all of my remarks.